# 3DGS-Drag: Dragging Gaussians for Intuitive Point-Based 3D Editing

**Jiahua Dong**    **Yu-Xiong Wang**
University of Illinois Urbana-Champaign
{jiahuad2, yxw}@illinois.edu

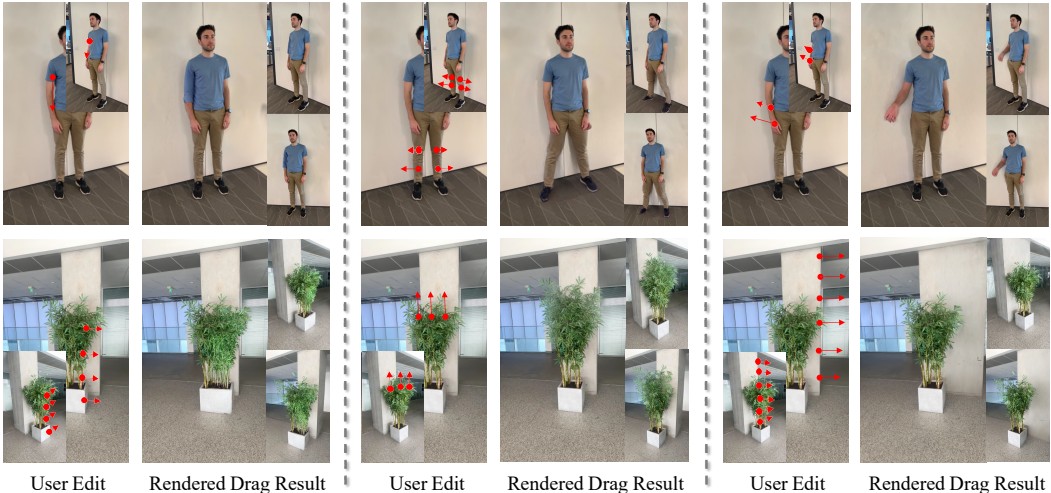

User Edit    Rendered Drag Result    User Edit    Rendered Drag Result    User Edit    Rendered Drag Result

Figure 1: **Our proposed 3DGS-Drag framework enables high-quality 3D drag editing:** Users only need to input 3D handle points (circle) and target points (triangle). Our method precisely moves the handle points to match the target points while preserving the overall content and details.

## ABSTRACT

The transformative potential of 3D content creation has been progressively unlocked through advancements in generative models. Recently, intuitive drag editing with geometric changes has attracted significant attention in 2D editing yet remains challenging for 3D scenes. In this paper, we introduce 3DGS-Drag – a point-based 3D editing framework that provides efficient, intuitive drag manipulation of real 3D scenes. Our approach bridges the gap between deformation-based and 2D-editing-based 3D editing methods, addressing their limitations to geometry-related content editing. We leverage two key innovations: *deformation guidance* utilizing 3D Gaussian Splatting for consistent geometric modifications and *diffusion guidance* for content correction and visual quality enhancement. A *progressive* editing strategy further supports aggressive 3D drag edits. Our method enables a wide range of edits, including motion change, shape adjustment, inpainting, and content extension. Experimental results demonstrate the effectiveness of 3DGS-Drag in various scenes, achieving state-of-the-art performance in geometry-related 3D content editing. Notably, the editing is efficient, taking 10 to 20 minutes on a single RTX 4090 GPU. Our code is available at https://github.com/Dongjiahua/3DGS-Drag.

## 1 INTRODUCTION

Recent years have witnessed remarkable advancements in 3D scene representation techniques, such as Neural Radiance Fields (NeRF) (Mildenhall et al., 2021) and 3D Gaussian Splatting (3DGS) (Kerbl et al., 2023). These methods have revolutionized the way we capture, represent, and synthesize 3D content, offering unprecedented levels of detail and realism. Inspired by their success

and the blooming development of 2D generative models (Rombach et al., 2022), recent works in 3D generation (Tang et al., 2024; Poole et al., 2023) can now generate 3D content with high quality and efficiency. However, precise and intuitive editing of 3D scenes remains a challenge, particularly in contrast to the sophisticated editing capabilities available for 2D images. While 2D editing methods like DragGAN (Pan et al., 2023) offer point-based manipulation, extending such functionalities to 3D scenes presents substantial technical hurdles.

Specifically, the underexplored capability behind is to achieve *intuitive content editing* with *geometric change*. The recent progress in 3D editing can be roughly grouped into two classes: deformation-based and 2D-editing-based. The deformation-based methods (Huang et al., 2024; Xie et al., 2024) primarily focus on motion editing, assuming strong geometry prior (Xie et al., 2024) or relying on video to learn motion pattern (Huang et al., 2024). Besides the requirement for sufficient prior information, they naturally cannot intuitively edit unseen content. For 2D-editing-based methods, recent works (Haque et al., 2023; Dong & Wang, 2023; Chen et al., 2024a) have attempted to distill the editing ability from 2D diffusion models (Brooks et al., 2023) by editing the dataset of different view images with the 2D diffusion model. These approaches remain limited to appearance modifications and minor geometric adjustments, since larger 2D geometric edits fail to converge to 3D. The text guidance they used also sometimes causes incorrect edits, because the diffusion model fails to understand the text prompt. *Bridging the geometric editing ability from deformation and the content editing ability from 2D-editing models has not yet been well studied.*

Motivated by these observations, we introduce 3DGS-Drag – *an intuitive 3D drag editing method for real scenes*. Extending the flexible editing format of DragGAN (Pan et al., 2023), we take 3D handle points and target points as inputs, aiming for geometry-related 3D content editing. Our key insight is to fully leverage two sources of guidance for 3D content editing, which explicitly regularize the edits from different views to be consistent and optimized toward the target 3D points. The first guidance is *deformation guidance*. Benefiting from the explicit representation of 3D Gaussian Splitting (Kerbl et al., 2023), we propose a simple but effective deformation strategy without the requirement for prior information. With such a strategy, we directly deform the 3D Gaussians and leverage them as guidance for different views. Moreover, the deformation of the Gaussians facilitates optimization around the deformed space, simplifying the generation of detailed geometry. The second one is *diffusion guidance*. Notably, since there is no prior information in our setting, the deformed Gaussians always have incorrect content and artifacts. We use the diffusion model to correct the content and improve the visual quality. This guidance is grounded in our observation that a fine-tuned diffusion model serves as a view-consistent editor for a 3D scene. Consequently, it achieves better consistency given previous deformation guidance.

To support more challenging 3D drag edits, we further propose a *progressive* editing strategy. Specifically, we divide the drag operation into several intervals and proceed with editing step by step. The continuity of editing is guaranteed by a 3D relocation strategy. In the end, our experimental results demonstrate the effectiveness of our 3DGS-Drag in various scenes and editing. We resolve the challenges of a 3D drag operation and indicate an enhancement in multi-view consistency compared to prior techniques.

Our major **contributions** can be summarized as follows: 1) We present a novel framework for editing 3D scenes, featuring a point-based drag editing approach. 2) We propose an effective method to bridge 3D deformation guidance and diffusion guidance for conducting geometry-related 3D content editing. 3) We further propose a progressive drag editing method to improve editing results. 4) Extensive evaluations show our method achieves state-of-the-art results in such setting, which implicitly includes motion change, shape adjustment, inpainting, and content extension.

## 2    RELATED WORK

### 2.1    2D IMAGE EDITING

Initially, the image generation methods rise from generative adversarial networks (GAN) (Goodfellow et al., 2014; Karras et al., 2019). Based on its latent representation, early works tried to modify the latent to adjust certain attributes or contents of the image (Abdal et al., 2021; Endo, 2022; Härkönen et al., 2020; Leimkühler & Drettakis, 2021). However, due to the limited capability of the GAN model and the implicit representation of the latent code, it is hard to achieve high-

quality and detailed edits. Recently, diffusion models have shown great potential for text-to-image tasks (Rombach et al., 2022). Its feature map representation and the large-scale data empower lots of image editing methods (Kawar et al., 2023; Ramesh et al., 2022; Meng et al., 2022; Brooks et al., 2023). SDEdit (Meng et al., 2022) performs a nosing and denoising procedure to keep the structural information and change the details. Instruct-Pix2Pix (Brooks et al., 2023) builds an instruction editing dataset and train the diffusion model to edit the image following the instruction. Compared with previous methods, Instruct-Pix2Pix shows better editing consistency.

Although text-based image editing can generate high-fidelity results, it cannot reach fine-grained editing. DragGAN (Pan et al., 2023) proposed a point-based interactive editing method. The user inputs several handle points and target points; then, the latent will be optimized to move the handle points to the target. To improve the generality, DragDiffusion (Shi et al., 2024) transfers this technique to diffusion models (Rombach et al., 2022). Later, SDE-Drag (Nie et al., 2024) and RegionDrag (Lu et al., 2024) further improve the performance. An inverse-forward process is necessary for these diffusion-based methods, making this operation time-consuming. In addition, there is no 3D consistency guaranteed in such 2D models, thus not available to be directly applied to 3D.

In this paper, we adopt the 2D diffusion model to perform 3D-consistent view correction. Our editing not only generates intuitive new content but also removes potential 3D artifacts.

## 2.2 2D-EDITING-BASED 3D EDITING

Previous to 3DGS (Kerbl et al., 2023), Neural Radiance Fields (NeRF) (Mildenhall et al., 2021) is used as a common connector from 3D representation to 2D models. Early works on NeRF can only deal with color and shape adjustment (Chiang et al., 2022; Huang et al., 2021; 2022; Wu et al., 2023; Bao et al., 2023; Zhang et al., 2022; Jambon et al., 2023). SNeRF (Nguyen-Phuoc et al., 2022) proposes to use an image stylization model, achieving high-quality stylization results. Later, NeRF-Art (Wang et al., 2023) uses CLIP (Radford et al., 2021) to distill the knowledge to NeRF. However, since the CLIP is not a generative model and is highly semantic-based, such an approach cannot get results with high fidelity. Instruct-NeRF2NeRF (Haque et al., 2023) proposes to use the Instruct-Pix2Pix model to Iteratively edit the dataset. They can edit various scenes with a broad range of instructions. ViCA-NeRF (Dong & Wang, 2023) proposes to directly edit the dataset without fine-tuning NeRF. Specifically, they make multi-view consistent edits by utilizing the depth of information. DreamEditor (Zhuang et al., 2023) proposes to use a fine-tuned Dreambooth (Ruiz et al., 2023) to help with editing. ConsistentDreamer (Chen et al., 2024a) further fine-tune a ControlNet to give more detailed edits. However, all these methods are limited by the 3D consistency from different views, thus only available to make subtle geometric changes. PDS (Koo et al., 2024) propose a new distillation loss to help improve the result but suffer from degeneration of rendering quality and the ability for sufficient geometric editing.

Inspired by the efficiency of 3DGS, recent approaches (Fang et al., 2024; Chen et al., 2024b; Chen & Wang, 2024) propose to migrate the success of NeRF editing to 3G Gaussians. However, they are mainly following the idea of Instrcut-NeRF2NeRF (Haque et al., 2023) by changing the 3D representation, thus having similar limitations. Some approaches (Xie et al., 2023; Shen et al., 2024; Yoo et al., 2024; Dong et al., 2024) have attempted to extend the drag operation to 3D; however, they are limited to handling single objects. In contrast, our approach leverages the explicit representation of 3DGS and focuses on real scenes.

## 2.3 DEFORMATION-BASED 3D EDTING

3D deformation is a challenging task since the target is to generate unseen motions. Traditional methods (Sorkine-Hornung & Alexa, 2007; Sorkine, 2005) apply certain Laplacian coordinates for mesh deformation. Recently, people have focused on deformation in 3D representations like NeRF and 3DGS. Specifically, Xu & Harada (2022) proposes to build 3D cages as the motion prior to guide deformation. Yuan et al. (2022) reconstruct the mesh from NeRF and deform the mesh instead. NeuralEditor (Chen et al., 2023) requires dense point cloud deformation as input and applies point-like NeRF structure for deformation. All these methods need strong geometry prior to editing, which is hard and inconvenient in practice. PhysGaussian (Xie et al., 2024) considers Gaussian ellipsoids as a Continuum and integrates physics. SC-GS (Huang et al., 2024) samples control points as a structure-representing graph to guide motion. However, the physics simulation and continuum

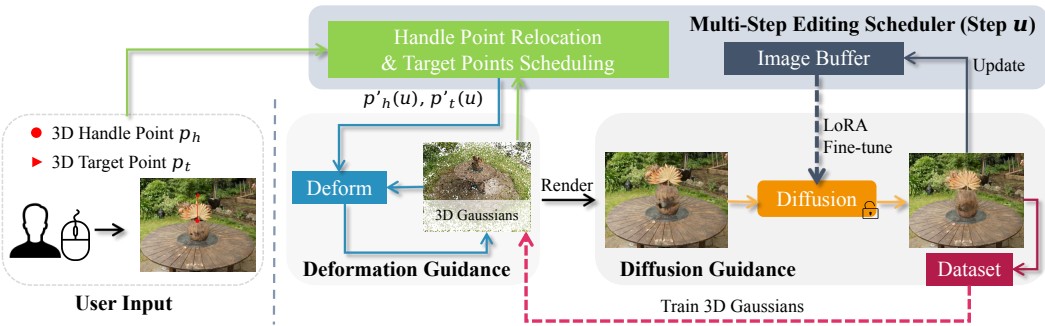

Figure 2: **Overview of 3DGS-Drag :** Given a trained 3D Gaussian splatting model and the dataset, we use the multi-step editing scheduler to calculate the intermediate handle points $p'_h(i)$ and target points $p'_t(i)$ for step $i$. In each step, we first deform the 3D Gaussians using handle points and target points. Then, we render the image for each view and correct it with a diffusion model. The final corrected images will be used to train 3D Gaussians to improve quality. The diffusion model is fine-tuned with LoRA for more consistent edits.

assumption make PhysGaussian less flexible and limited to continuous scenes. SC-GS's control points are an approximation of dense points, thus also relying on sufficient capture of the object's geometry. It also takes dynamic scenes as input to build prior motion knowledge.

The hard prior knowledge requirements or strict assumptions make these methods not suitable for large real scenes, where there is often only part-view information and complex layouts. In addition, they do not have the ability to create new parts. Rather than building a better deformation method, we propose a simpler deformation strategy for 3DGS to give rough deformations. Since the 2D generative models (Rombach et al., 2022) *already have the sense of normal motions and contents*, we borrow such knowledge to provide more flexible 3D edits

## 3 METHOD

### 3.1 PRELIMINARY

**3D Gaussian Splatting.** 3D Gaussian splatting (Kerbl et al., 2023) uses a collection of 3D Gaussians to represent 3D information, demonstrating effectiveness in object and scene reconstruction tasks. Each Gaussian is characterized by a center $\mu \in \mathbb{R}^3$, a scaling factor $s \in \mathbb{R}^3$, and a rotation quaternion $q \in \mathbb{R}^4$. The model also incorporates an opacity value $\alpha \in \mathbb{R}$ and a color feature $c \in \mathbb{R}^d$ for volumetric rendering, where $d$ indicates the degrees of freedom. The full set of parameters is denoted as $\Gamma$, where $\Gamma^i = \{\mu^i, s^i, q^i, \alpha^i, c^i\}$ represents the parameters for the $i$-th Gaussian.

### 3.2 FRAMEWORK OVERVIEW

Our framework is illustrated in Figure 2. It takes a pretrained 3D Gaussian splatting model and several handle points along with their corresponding target points as input. Specifically, the handle points are denoted as $p_h^{n\times3}$, and the target points are denoted as $p_t^{n\times3}$, where $n$ is the number of handle points. We aim to move the handle part to the target position while preserving similar content. Depending on the input points, this process may entail appearance and geometric changes, allowing more challenging edits with user-friendly inputs.

Different from the idea of 2D drag editing techniques (Pan et al., 2023; Shi et al., 2024; Lu et al., 2024), which either optimize or operate the inverse feature of a 2D image, we use *deformation-based geometric guidance* and *diffusion-based appearance guidance* for 3D editing. For a single step of drag operation, we first deform the 3D Gaussians with the provided handle and target points (Sec. 3.3). Such deformation is conducted in a copy-and-paste manner to allow more editing flexibility. Due to the sparsity and long-distance challenge of the drag operation, the rendering result from the deformed Gaussians have poor visual quality and incorrect content. Thus, we propose to use diffusion-guided image correction on the rendered images (Sec. 3.4), which efficiently corrects the contents and removes artifacts. To resolve editing with more aggressive changes, we propose

a multi-step editing scheduler to progressively edit the scene (Sec. 3.5). As the whole process is divided into intervals, the user can stop at any intermediate step when achieving a satisfactory outcome.

### 3.3 Deformation Guidance for Geometric Modification

As we aim to deform the 3D scenes to provide geometry guidance, we leverage 3DGS to benefit from its explicit representations and efficiency. The deformation involves two challenges in our task: (1) How to approximately deform the 3D Gaussians given sparse handle points and long-distance drag target, without structural modification to standard 3DGS; (2) How to avoid degeneration to direct deformation, allowing more flexibility to edits like moving, extending, and others. The proposed solution is described as follows. As a result, we achieve reliable deformation to 3DGS given limited point information.

**Drag Deformation.** The explicit representation of 3DGS enables efficient 3D deformation and adjustment. However, the real deformation function cannot be precisely computed, given only handle points and target points. Thus, we approximate it to give a rough geometry guidance. For the $i$th handle point $p_h^i$, we assign the Gaussians $P_h^i$ within a certain distance $\tau$ in 3D to this point. These Gaussians are considered to be deformed and guided by this handle point. The union of $\{P_h^i | i = 1, 2, ..., n\}$ is denoted as $P_h = \bigcup_{i=1}^n P_h^i$.

Firstly, we calculate the translation and rotation for each handle point. For the translation, we simply calculate it as: $\Delta p_h^i = p_t^i - p_h^i$. For the rotation, it is not to further change the position of handle points but to represent the potential orientation change. Since the 3D Gaussians are also parameterized by rotation $q$, such a parameter is crucial to guide the Gaussian deformation. However, our handle points are just coordinates without information on the orientation. To approximate the rotation, we calculate its relative rotation with its top-$K$ ($K = 2$) nearest handle points $\{p_h^k | k \in N_h^i\}$ where $N_h^i$ are the indices of top-$K$ nearest handle points. Linear weight is used due to the sparsity of the points. Specifically, the weight is calculated as:

$$w_h^{ik} = 1 - \frac{\left\| p_h^i - p_h^k \right\|_2^2}{\sum_{j \in N_h^i} \left\| p_h^i - p_h^j \right\|_2^2}. \tag{1}$$

Then, we calculate the relative rotation quaternion $\Delta q_h^{ik}$ between $p_h^i$ and $p_h^k$ (Details in Sec. D), and the quaternion $\Delta q_h^i$ of pair $(p_h^i, p_t^i)$ is calculated as $\Delta q_h^i = \sum_{k \in N_i} w_h^{ik} \Delta q_h^{ik}$.

After calculating each handle point's translation and rotation quaternion, we can interpolate the entire 3D Gaussians' deformation. Specifically for each Gaussian $\Gamma^i \in P_h$, the deformed Gaussian is interpolated from the transformation of top-$K$ ($K = 2$) nearby handle points $\{p_h^j | j \in N_i\}$ where $N_i$ are the indices of top-$K$ nearest handle points. The deformed center $\mu_d^i$ and rotation quaternion $q_d^i$ are:

$$w^{ik} = 1 - \frac{\left\| \mu^i - p_h^k \right\|_2^2}{\sum_{j \in N_i} \left\| \mu^i - p_h^j \right\|_2^2}, \tag{2}$$

$$\mu_d^i = \mu^i + \sum_{k \in N_i} w^{ik} \Delta p_h^k, \tag{3}$$

$$q_d^i = \sum_{k \in N_i} (w^{ik} \Delta q_h^k) \otimes q^i, \tag{4}$$

where $\mu^i$ and $q^i$ are the original center and rotation quaternion. $\otimes$ is the quaternion production. When there is only one handle point, no quaternion change will be applied. We do not directly update the old Gaussians to the deformed Gaussians since this limits deformation and is not suitable for tasks like "make his sleeves longer." Inspired by SDE-Drag (Nie et al., 2024), we use a copy-and-paste manner to place the deformed Gaussians and keep the old ones. To offer more flexibility for optimization, we adjust the opacity of the original Gaussians $P_h$ to a smaller value, allowing the 2D updates to determine whether to keep or remove the Gaussians.

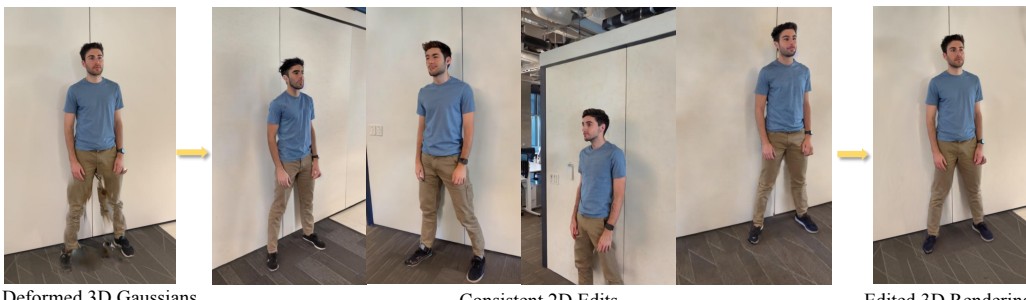

Deformed 3D Gaussians         Consistent 2D Edits         Edited 3D Rendering

Figure 3: **Multi-view consistent 2D edits:** With the deformed rendering as input, the fine-tuned diffusion model can perform multi-view consistent edits, and the artifacts and incorrect parts (shoes) are fixed.

**Local Editing Mask.** Since drag operations mainly focus on a part of the entire scene, local editing is necessary to maintain the background information. Following Gaussian Editor (Chen et al., 2024b), we assign a mask $M$ to the Gaussians of $P_h$, which are considered changeable. Different from Gaussian Editor, our work builds *both 3D and 2D local editing masks* to work with more complex scenes and geometry edits. For the 3D mask, we inherit the mask from the original Gaussians when deforming new Gaussians or during the densification procedure. These Gaussians outside of the mask are not changed in the optimization. For the 2D mask, we render the mask for each view and round it to (0, 1) with a threshold, resulting in masks $\{m^v\}$ where $v$ denotes the $v$th view. Note that the mask rendering is after the deformation, so the original region and the target region will both be covered. The mask is further dilated to change the context of the nearby area.

## 3.4 DIFFUSION GUIDANCE FOR APPEARANCE CORRECTION

The direct deformation of Gaussians often creates notable artifacts and cannot generate correct semantic content. Inspired by recent successes in 3D editing (Haque et al., 2023), we update the dataset to edit 3D scenes. However, integrating the concept of 2D dragging into a 3D context is non-trivial. Previous 2D drag methods often necessitate a time-consuming forward and backward process (Shi et al., 2024; Nie et al., 2024). Moreover, during the training process, the inconsistent 2D edits from different views make the final result deviate from expectations and full of artifacts. To address these issues, we propose to use inverse-free 2D image editing that achieves *stronger 3D consistency, efficiency, and quality*, relying on consistent renderings from the deformed 3D content. As shown in Figure 3, our method generates multi-view consistent 2D edits. In detail, given the rendered image from the deformed 3D Gaussians, we introduce an *Image2Image view correction* to obtain corrected 2D edits. To overcome the challenge of dataset editing with geometry change, we update the dataset in an *annealed dataset editing* way.

**Image2Image View Correction.** Although the deformed Gaussian gives better 3D consistency, it cannot benefit from latent-based drag methods (Pan et al., 2023). This is because the 3D consistency is ensured with newly rendered images. In contrast, latent-based methods heavily rely on operating the feature map of the same image. Inspired by the common approach for image editing (Meng et al., 2022), we add noise and then denoise it through the Dreambooth (Ruiz et al., 2023) model. By changing the image to a sketch level and denoising it, the diffusion model can partially understand and complete the deformed part.

To mitigate the influence of randomness from the diffusion model, the Dreambooth model is fine-tuned on each scene with LoRA (Hu et al., 2022). We find that after fine-tuning, the diffusion model becomes a multi-view consistent editor. The experiment results in Figure 3 show that the diffusion model can successfully understand the deformed image and generate an image with the correct content even without the inverse process. However, such corrections still cannot fully converge in one update, requiring a better dataset editing strategy as follows.

**Annealed Dataset Editing.** Iterative dataset editing has been a common approach for 3D appearance editing (Haque et al., 2023). The idea is to progressively change the appearance of 3D and use the rendering to guide consistent 2D editing further. However, such a strategy does not work well with geometry-related edits because it is harder to converge given inconsistent geometry. In addition,

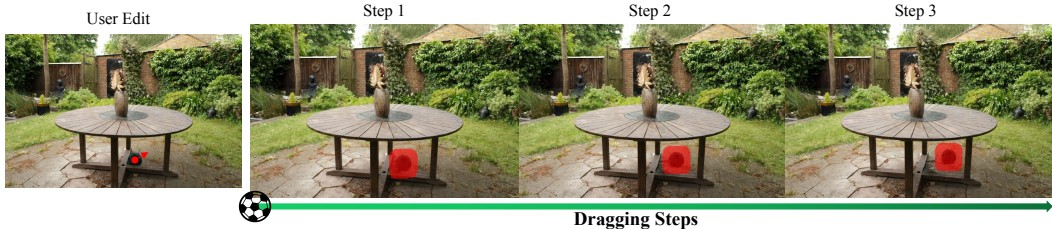

Figure 4: **Intermediate dragging steps and tracked mask:** Our method conducts progressive editing toward the target point. The dragged Gaussians are tracked to achieve aggressive edits.

long-term iterative updates also accumulate serious blurriness (Haque et al., 2023). To address this, we propose to update the dataset with limited $A$ times, and each time anneals the strength (Meng et al., 2022) for Image2Image view correction. The annealing function is as follows:

$$S(a) = S_{\text{init}} - \frac{a-1}{A}(S_{\text{init}} - S_{\text{final}}), \quad a = 1, 2, 3, ..., A, \tag{5}$$

where $S_{\text{init}}$ and $S_{\text{final}}$ are the initial strength and final strength respectively. $S(a)$ denotes the strength for the $a$th updates. Note that lower strength means that diffusion starts from later timesteps, resulting in finer detail correction. Our strategy performs editing in a coarse-to-fine manner. Each time, all the views are updated to prevent accumulated errors.

**Loss Function.** With the rendered image $I_r^v$ from 3D Gaussians, the corresponding edited image $I_e^v$ as the editing area's groundtruth, the original image $I_o^v$ as background groundtruth and mask $m^v$ for view $v$, our loss function for training 3D Gaussians is formulated as:

$$\mathcal{L} = \sum_{v=1}^{V}(\lambda_1\mathcal{L}_1(I_r^v, I_o^v) + \lambda_{\text{ssim}}\mathcal{L}_{\text{ssim}}(I_r^v, I_o^v)) \odot (1 - m^v) + \lambda_{\text{lpips}}\mathcal{L}_{\text{lpips}}(I_r^v, I_e^v) \odot m^v, \tag{6}$$

where $\mathcal{L}_1$ and $\mathcal{L}_{\text{ssim}}$ are to ensure local editing. $\mathcal{L}_{\text{lpips}}$ is the LPIPS (Zhang et al., 2018) loss function to correct the editing area. $\lambda_1$, $\lambda_{\text{ssim}}$ and $\lambda_{\text{lpips}}$ are the weighting coefficient for each loss.

### 3.5 FROM ONE-STEP TO MULTI-STEP DRAG EDITING

The previous sections introduce the one-step drag editing using our method. As the long-distance drag operation often requires more than one step to avoid corruption, we propose a multi-step editing scheduler to solve such problems. Specifically, we split the drag operation into $T$ intervals and set the progressive target points $\{p_t'(u)|u = 1, 2, ..., T\}$. In each interval, we perform drag toward the corresponding target points:

$$p_t'(u) = p_h + \frac{u}{T}(p_t - p_h), \tag{7}$$

However, the actual handle point position usually changes when training 3D Gaussians. We propose relocating the handle points at the end of every interval to make the next interval's deformation more precise. In addition, we further conduct history-aware diffusion fine-tuning to improve the ability for more aggressive editing.

**Handle Point Relocation.** The handle point relocation is performed after each interval's training process. To keep track of the handle points, we use the Gaussians associated with each handle point. Specifically for handle point $p_h^i$, we update it with the averaged position change of Gaussians $P_h^i$. As shown in Figure 4, the dragged part can be successfully relocated. Note that the assigned Gaussians $P_h^i$ are updated to newly deformed Gaussians during deformation and inherited from parents during the densification process of training. The local mask is updated as the union with the mask.

**History-Aware Diffusion Fine-Tuning.** For long-distance drag operations, the edited 2D images can shift out of the diffusion model's domain since it is fine-tuned on the original images, resulting in degeneration back to the original images. We build an image buffer to fine-tune the diffusion model. The diffusion model will be fine-tuned with the image buffer every interval. Initially, the buffer only contains original images, and the newly edited result will be added to the buffer during intervals.

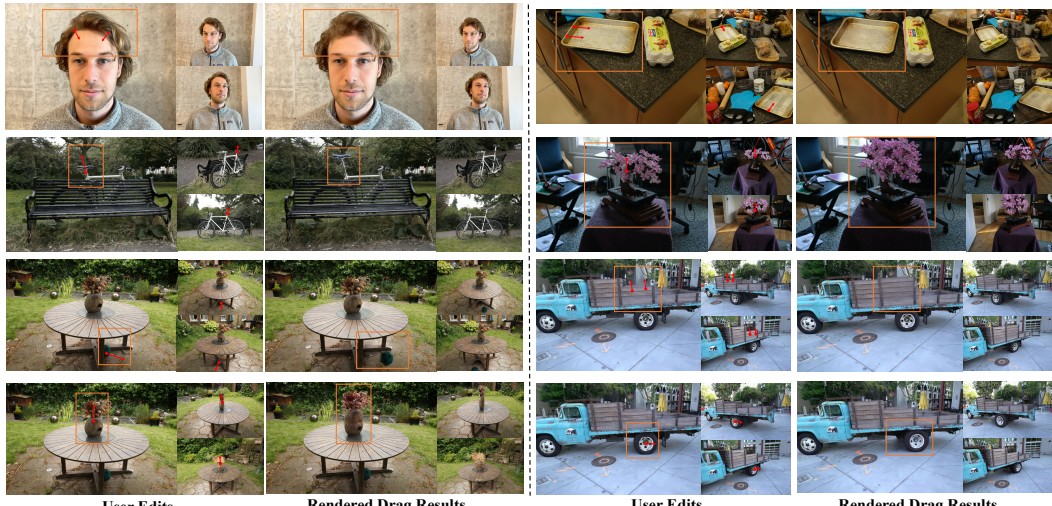

Figure 5: **Qualitative results in various scenes**: Our method can handle complex scenes and generate highly detailed results. With a simple drag input, 3DGS-Drag can identify the 3D context and perform edits like moving objects, inpainting the background, adjusting appearance, modifying object shape, and adjusting motion. The orange bounding boxes highlight the modified regions.

## 4  EXPERIMENT

### 4.1  IMPLEMENTATION DETAILS

**User Input.** Our user input is one or multiple handle points and corresponding target points. The input points are in 3D space. The user can specify the sphere radius of that handle point to adjust the editing scale. We automatically perform local editing by applying the mask rendered from assigned Gaussians. The mask is dilated to change the necessary context.

**Drag Editing.** The pretrained 3D Gaussians are trained with original 3D Gaussian Splatting (Kerbl et al., 2023). During editing, 50 views are selected to enable efficient editing by default. Specifically, we choose the views with a larger visible area on the handle points' Gaussians, which is determined by the local editing mask on each view. We fine-tune the Dreambooth model (Ruiz et al., 2023) with LoRA (Hu et al., 2022). Initially, it is fine-tuned on the selected views. We use batch size 4 and train for 200 iterations. After each dragging step, we continue fine-tuning the diffusion model for 50 iterations with the updated image buffer in each interval. Each time, the newly edited image will be enqueued. The loss weight of $\lambda_1$, $\lambda_{\text{ssim}}$ and $\lambda_{\text{lpips}}$ are set to $8, 2, 1$ respectively. Note that the $\lambda_1$ and $\lambda_{\text{ssim}}$ are 10 times bigger than normal to ensure the background.

**Datasets.** our experiments include edits on eight scenes, using the published datasets from Instruct-NeRF2NeRF Haque et al. (2023), PDS Koo et al. (2024), Mip-NeRF360 Barron et al. (2022), and Tank and Temple Knapitsch et al. (2017).

### 4.2  QUALITATIVE EVALUATION

**Editing Results in Various Scenes.** We show editing results from different views in Figure 1 and Figure 5. Since the handle and target points are in 3D, we plot them in 2D for illustration. Each drag is represented by a red arrow where the start is the handle point, and the end is the target point. In the standing-person scene in Figure 1, when we raise one hand, this is very challenging since the arm is only observed partially, and the part under the arm is unknown. Our method also shows the ability to generate new poses and fix the texture on the pants below the arm. We are also able to change the leg motion and extend the sleeves. When dealing with more complex scenes, such as the bamboo scene in Figure 1, 3DGS-Drag can understand the texture of the plant and extend it to be taller or wider. We can also easily change part of the background, like the wall. When the drag operation is to move the football, we can separate this object from the background and inpaint the texture at the original position instead of an empty region. In short, our drag operation can understand different operations

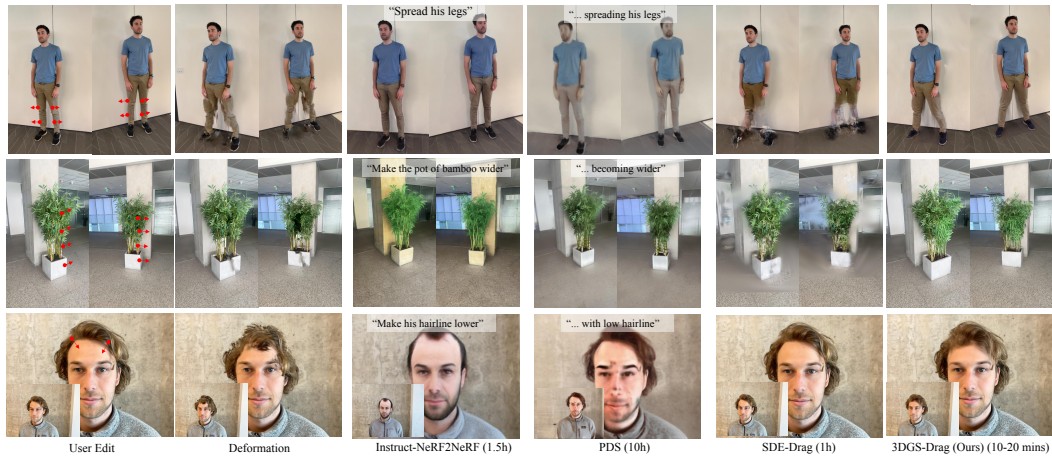

User Edit     Deformation     Instruct-NeRF2NeRF (1.5h)     PDS (10h)     SDE-Drag (1h)     3DGS-Drag (Ours) (10-20 mins)

Figure 6: **Baseline comparisons:** Compared with baselines, 3DGS-Drag achieves high-quality, fine-grained editing by correctly modifying different parts and in terms of efficiency. Specifically, Instruct-NeRF2NeRF (Haque et al., 2023) and PDS (Koo et al., 2024) cannot correctly edit. Deformation results in incomplete edits, and SDE-Drag (Nie et al., 2024) sometimes fails to make changes.

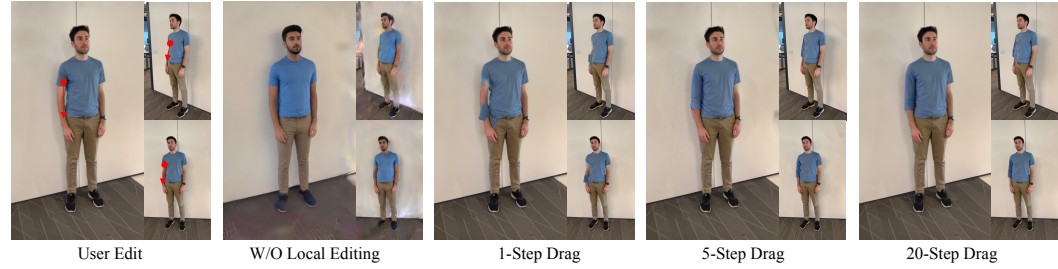

User Edit     W/O Local Editing     1-Step Drag     5-Step Drag     20-Step Drag

Figure 7: **Ablation study on the local mask and drag steps:** Without the local mask, the scene will be blurred, resulting in failed edits. Using very few steps makes it hard to achieve aggressive edits. More steps will slightly improve the performance.

in front-view or 360-degree scenes, such as moving objects and extending objects, demonstrating the ability to identify the 3D context.

**Baseline Comparison.** Since there is no directly comparable work on intuitive 3D drag operation in real scenes, we extend and re-purpose representative baselines. The results are shown in Figure 6. Specifically, the comparison with baselines is listed as follows:

- *Instruct-NeRF2NeRF* (Haque et al., 2023): We manually create text descriptions for drag operations in this baseline. Then, we use Instruct-NeRF2NeRF to edit the scene. The model fails to give edits for the 'person' scene. For the more complex 'garden' scene, Instruct-NeRF2NeRF just blurs the rendering. This demonstrates its insufficient ability to perform geometric modification.
- *Deformation*: We use our deformation to represent the previous deformation-based approaches since we have different input settings. Notably, the geometry is moved, which results in a lot of incorrect content and artifacts.
- *PDS*: PDS (Koo et al., 2024) claims to be able to change the geometry, but this method struggles in all three editing scenarios. In addition, PDS tends to create noisy and blurred editing results compared with others.
- *SDE-Drag*: One alternative solution is to simply use the 2D drag method on each view. Here we choose SDE-Drag (Nie et al., 2024) in comparison. However, such a strategy cannot reach consistent edits, resulting in flawed results or failure cases in editing.

Compared with these baselines, our methods achieve significantly better editing results, with better details and correct content. Remarkably, for the "lower his hairline" text prompt, both Instruct-NeRF2NeRF and PDS misunderstand the text and make the hairline higher, which further emphasizes the importance of intuitive 3D editing.

**Ablation Study** The diffusion guidance's effectiveness is validated when compared with the deformation approach (Figure 6). Here, We further ablate the local mask and multi-step strategies in Figure 7. (1) When local editing is not applied, the entire scene is blurred, and the edits fail. This is due to the optimization issue: inconsistent edits will create large floats in 3D Gaussians. (2) For drag steps, we compare three different drag steps from $[1, 5, 20]$, finding that more or fewer steps lead to different insights. When using a single step, the deformed Gaussians cannot give enough guidance to the diffusion model, resulting in broken edits. Thus, one-step drag editing usually meets challenges when we have more aggressive edits. When applying more steps (20 steps), the editing quality is slightly improved. This illustrates that 3DGS-Drag is robust when updated more times. However, since more steps will slow the execution, choosing an appropriate number of steps is better.

### 4.3 QUANTITATIVE EVALUATION

Quantitatively evaluating 3D editing results is often challenging since there lacks ground truth. Here, we use two metrics for evaluation: user preference and GPT score, shown in Figure 8. For user preference, we conducted a user study across 19 subjects and collected their preference for each edit. For the GPT score, since GPT with vision has been proven to be a human-aligned evaluator (Wu et al., 2024), we use gpt4-o to evaluate each editing, rating in 5 levels. Specifically, we measure (1) whether the content is correctly edited and (2) the rendered image quality for each method. Our method achieves the best results on all these metrics.

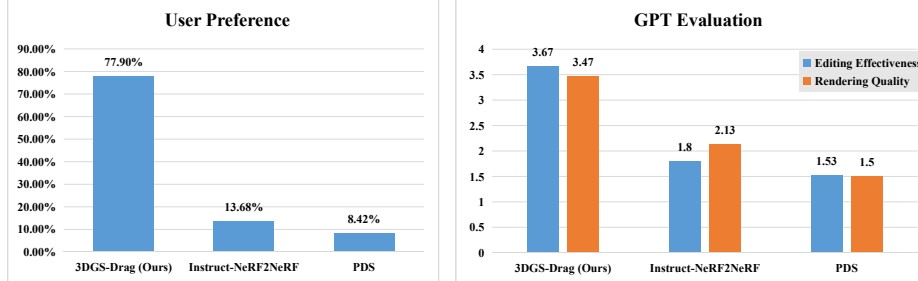

Figure 8: **Quantitative evaluation:** We conduct both user study and GPT evaluation on the editing results. Compared with Instruct-NeRF2NeRF (Haque et al., 2023) and PDS (Koo et al., 2024), 3DGS-Drag performs significantly better.

### 4.4 DISCUSSION

**Limitations.** Similar to previous diffusion-based 3D editing methods (Chen et al., 2024b; Haque et al., 2023), our approach relies on the diffusion model to provide accurate guidance. Thus, our method may yield suboptimal results when the target object is too small within the field of view or when the scene exhibits considerable size and complexity. We also cannot deal with drag operations that are too aggressive. In such cases, the object may be relocated to areas with restricted visibility, which is out of vision for most views.

**Running Time.** When using 50 views for editing, our method needs 15 minutes. Specifically, about 2 minutes are needed for initial diffusion model fine-tuning, and 13 minutes are needed for the rest of the editing process. In comparison, Instruct-NeRF2NeRF (Haque et al., 2023) needs one hour. The running time is tested on a single RTX 4090 GPU.

### 5 CONCLUSION

In this paper, we introduced 3DGS-Drag , an intuitive drag editing approach for 3D scenes. In contrast to previous work (Haque et al., 2023; Dong & Wang, 2023; Wang et al., 2023), which mainly focuses on appearance, we address the challenge of geometry-related content editing. Empirical experiments show that our method can achieve highly detailed edits across various scenes. Such an advantage stems primarily from our two key contributions: the copy-and-paste Gaussian deformation and the diffusion correction. We showcase that our method enables previously challenging edits, paving the way for exploring new possibilities in 3D editing.

ACKNOWLEDGMENTS

This work was supported in part by NSF Grant 2106825, NIFA Award 2020-67021-32799, the Toyota Research Institute, the IBM-Illinois Discovery Accelerator Institute, the Amazon-Illinois Center on AI for Interactive Conversational Experiences, Snap Inc., and the Jump ARCHES endowment through the Health Care Engineering Systems Center at Illinois and the OSF Foundation. This work used computational resources, including the NCSA Delta and DeltaAI supercomputers through allocations CIS220014 and CIS230012 from the Advanced Cyberinfrastructure Coordination Ecosystem: Services & Support (ACCESS) program, as well as the TACC Frontera supercomputer, Amazon Web Services (AWS), and OpenAI API through the National Artificial Intelligence Research Resource (NAIRR) Pilot.

REPRODUCIBILITY STATEMENT

Our code is released at https://github.com/Dongjiahua/3DGS-Drag. For the implementation details, we have covered our mathematical details in Sec. 3.3 and training details in Sec. 4.1. The framework architecture is fully introduced in Sec. 3. All the datasets we used are publicly available, as explained in Sec. 4.1.

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

## A    DEMO VIDEO

A **demo video** of our framework description and editing results is included in the supplementary material.

## B    ADDITIONAL EXPERIMENTS

### B.1    QUALITATIVE RESULTS ON LARGE OBJECT MOVEMENTS

We further conduct experiments on object movements, specifically including large movements of both regular-sized and large objects. As shown in Figure 9, our method succeeds in handling long-distance movements, such as repositioning the flowerpot. Furthermore, for very large objects like the truck and the table, our approach effectively moves them in a specified direction while minimizing artifacts in the removed and placed regions. These results showcase the generalizability of our method.

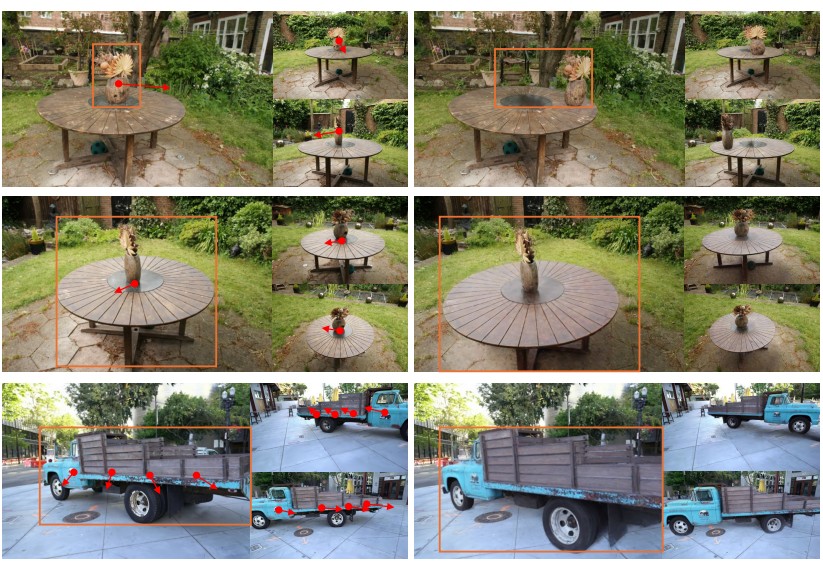

User Edits        Rendered Drag Results

Figure 9: **Additional qualitative results for larger movements and larger objects:** Our method succeeds in longer-range movements like moving the flowerpot and large object movements like moving the table.

### B.2    ABLATION ON DATASET EDITING STRATEGY

To validate the importance of our dataset editing strategy, we conduct a comparison between using our annealed dataset editing and the iterative dataset editing (Haque et al., 2023). As shown in Figure 10, editing one frame each time cannot change the geometry due to inconsistent constraints from other unedited views, leading to a degenerated result in the original scene. Instead, our method can successfully make the edits.

### B.3    QUANTITATIVE ABLATION ON LOCAL EDITING

To further demonstrate the effectiveness of our local editing, we conduct quantitative evaluation on the "extend sleeves" edit. Specifically, we calculate the similarity between the rendered edited result and the originally rendered image in the unedited pixels. As shown in Table 1, our local editing strategy demonstrates a strong capability to preserve the unedited regions and backgrounds effectively.

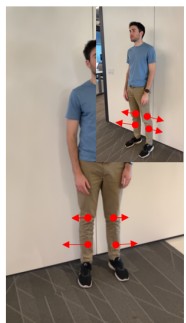 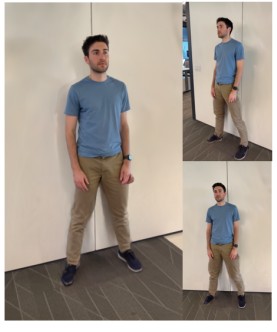 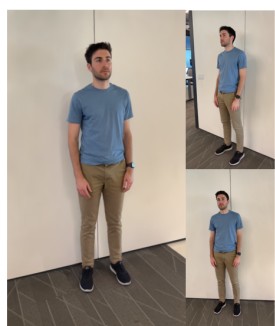

User Edit     Annealed Dataset Editing (Ours)     Instruct-NeRF2NeRF Dataset Editing

Figure 10: **Ablation on dataset editing strategies:** Iterative dataset editing from Instruct-NeRF2NeRF (Haque et al., 2023) leads to degenerated results. In contrast, our annealed dataset editing maintains the geometry change.

|  | SSIM↑ | PSNR↑ | LPIPS↓ |
|---|---|---|---|
| Local Editing | **0.995** | **43.43** | **0.004** |
| Non-Local Editing | 0.901 | 24.44 | 0.158 |

Table 1: **Quantitative ablation on local editing**: Our local editing strategy demonstrates a strong capability to preserve the unedited regions and backgrounds effectively.

## B.4 SCALED USER STUDY

To improve the generalizability of our user study, we increased the number of participants from 19 to 99. As shown in Figure 11, the key conclusion that our method surpasses previous baselines remains the same.

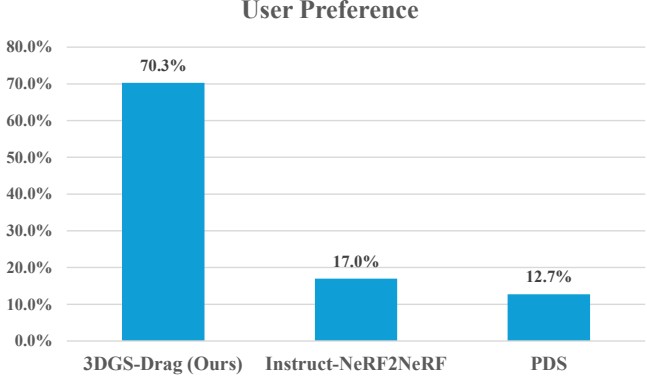

Figure 11: **Scaled user study with 99 participants:** Our method still achieves significantly better preference over the baselines.

## B.5 COMPARISON WITH 2D DRAG METHODS

We conduct qualitative comparisons on 2D drag edits to verify our method's effectiveness. Specifically, we focus on our deformation-guided diffusion editing quality, comparing the result with the previous 2D drag methods, Dragdiffusion (Shi et al., 2024) and SDE-Drag (Nie et al., 2024). As shown in Figure 12, DragDiffusion tends to create many more artifacts and unrelated textures, while our result is cleaner and matches better with the edit. Considering SDE-Drag (Nie et al., 2024), it fails to move the leg correctly and instead generates an object at the location. The results show that our method can achieve better consistency by directly operating at the image level.

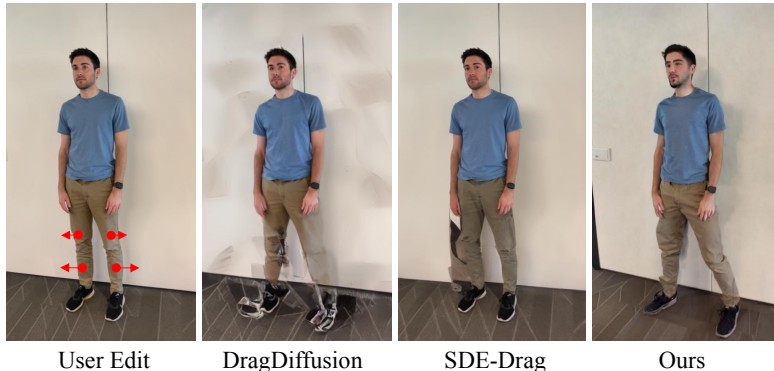

|  User Edit  |  DragDiffusion  |  SDE-Drag  |  Ours  |

Figure 12: **Comparison on 2D drag results:** Compared with recent 2D drag methods, our method does not need a time-consuming inverse-forward process and produces more consistent results. In comparison, the baseline DragDiffusion generates noisy legs and floor. SDE-Drag succeeds in maintaining the background but inserts objects in the hand and does not correctly move the leg.

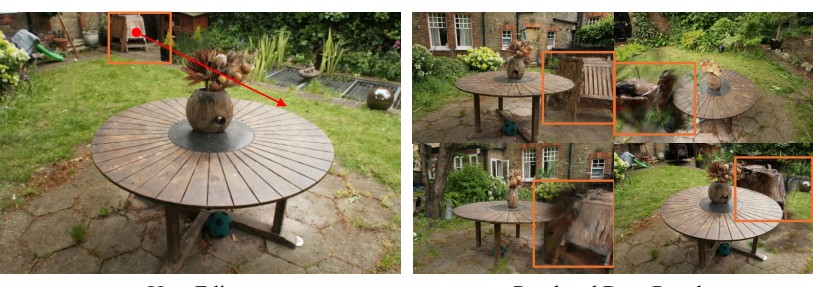

|  User Edits  |  Rendered Drag Results  |

Figure 13: **Limitation on generating unseen side:** When dragging background objects with an unseen side to the foreground, the results from other views are incorrect.

## C    IN-DEPTH DISCUSSION OF LIMITATIONS

Our method encounters two specific failure cases: generating the unseen side of an object and dragging objects outside the border area. Here, we present qualitative results to further elucidate these limitations.

### C.1    GENERATING UNSEEN SIDE

As shown in Figure 13, when moving the wooden support to the foreground, the unseen portions (e.g., the back of the object) are rendered incorrectly and exhibit noticeable artifacts. This occurs because there are no 3D Gassians to represent the unseen back side. Consequently, the deformed result contains meaningless patterns that cannot be corrected through the diffusion process.

### C.2    DRAGGING OBJECTS OUTSIDE THE BORDER

As shown in Figure 14, we struggle to refine the artifacts when only a portion of the object is visible at the border. This is because the diffusion model has less capability to correct the boundary part due to the ambiguity in interpolating the part to the whole. In addition, since it is observed by sparser views, it raises further challenges for optimizing 3D Gaussians.

### C.3    POTENTIAL BIASES IN QUANTITATIVE EVALUATIONS

While we employ both human preference scores and automated metrics from GPT evaluation, there could still be potential biases, such as those arising from the participant selection process or the specific version and training data of the GPT models used. This challenge is common in generative

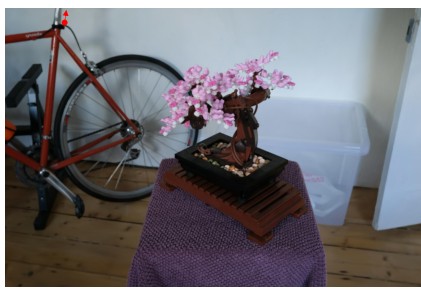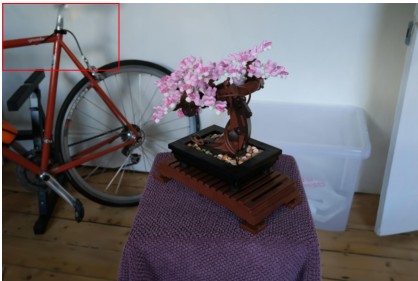

Figure 14: **Limitation on dragging objects outside the border:** Refining and optimizing become challenging when dragging towards the unseen or border area.

modeling, where there is no ground truth. Future work would benefit from conducting larger-scale human studies with more diverse participant pools and developing more comprehensive evaluation protocols that can better assess both geometric accuracy and visual fidelity of 3D edits.

## D  ADDITIONAL DEFORMATION DETAILS

The calculation of relative rotation $\Delta q_h^{ik}$ is briefly described as follows. Given handle points $p_h^i$ and $p_h^k$ and their corresponding target points $p_t^i$ and $p_t^k$ respectively, we firstly calculate the unit vectors as

$$v_h^{ik} = \frac{p_h^k - p_h^i}{\|p_h^k - p_h^i\|} \quad \text{and} \quad v_t^{ik} = \frac{p_t^k - p_t^i}{\|p_t^k - p_t^i\|} \tag{8}$$

Next, we compute the cross product and the dot product of these two unit vectors:

$$\mathbf{r} = v_h^{ik} \times v_t^{ik}, \quad s = v_h^{ik} \cdot v_t^{ik} \tag{9}$$

We then construct the quaternion $\Delta q_h^{ik}$ by combining the dot product and cross product:

$$\Delta q_h^{ik} = [s, \mathbf{r}_x, \mathbf{r}_y, \mathbf{r}_z] \tag{10}$$

Finally, we standardize and normalize the quaternion to ensure it has unit length.

## E  SOCIAL IMPACT AND FUTURE WORK

**Future Work.** For future work, we plan to extend current progressive editing capabilities to generate 3D animations. As 3DGS-Drag is able to move or modify objects progressively, it is possible to generate long-term trajectories and human motions. In addition, we will focus on improving the model's scalability to accommodate larger scenes with dynamic objects and shadow effects.

**Potential Social Impact.** The potential societal impact of 3DGS-Drag spans across multiple dimensions. Designed as a fine-grained editing model, our 3DGS-Drag offers convenient manipulation of 3D scenes and robust support for AR applications. In addition, given the rapid development and widespread adoption of 3D Gaussians, our method seamlessly integrates with this ecosystem. With its user-friendly interface requiring only the selection of handle and target points, our model is accessible even to untrained individuals.

