# OpenReview forum: "3DGS-Drag: Dragging Gaussians for Intuitive Point-Based 3D Editing"
_ICLR.cc/2025/Conference — ICLR 2025 Poster_

### Official Review · Reviewer_Yk2w · 2024-10-28

**Soundness:** 3
**Presentation:** 2
**Contribution:** 2
**Rating:** 6
**Confidence:** 3

**Summary:**

This work introduces a novel framework for point-based 3D content editing, addressing the limitations of existing 2D editing tools when applied to 3D scenes. The method enables users to intuitively edit 3D scenes by specifying 3D handle and target points, with the system automatically adjusting the geometry while preserving visual details. The approach combines two key innovations: deformation guidance, using 3D Gaussian splatting for geometric changes, and diffusion guidance for correcting content and enhancing visual quality. A multi-step editing strategy further ensures smooth and consistent results. Experiments show that 3DGS-Drag outperforms existing methods in various tasks, including motion changes, shape adjustments, and inpainting, achieving state-of-the-art results with significant efficiency, requiring only 10-20 minutes per edit on an RTX 4090 GPU.

**Strengths:**

The paper is well-written and easy to follow.

* The proposed multi-step progressive editing strategy enables higher-quality edits while maintaining detail and multi-view consistency, addressing limitations found in previous 3D Gaussian Splatting (3DGS)-based editing methods like GaussianEditor and viCA-NeRF.

* The framework uniquely integrates 3D Gaussian Splatting for geometric deformation with diffusion models for content correction. This combination helps maintain visual consistency across views and supports versatile editing capabilities.

* Through qualitative and quantitative comparisons with baseline methods, the paper demonstrates that the proposed method achieves superior results in terms of editing quality, user preference, and performance across different scenarios.

**Weaknesses:**

1. The study introduces user preference and GPT evaluations for quantitative assessment. However, it does not clearly address potential biases in the user study, such as participant selection. Additionally, the sample size of 19 subjects may be insufficient to support the evaluation's conclusions. Recognizing that quantitative evaluation of diffusion-based methods can be challenging, as is the case with GPT evaluations, it is important to acknowledge these limitations.

2. The qualitative figures do not clearly demonstrate the effectiveness of local editing. The proposed method appears to perform relatively well when editing small objects or areas. It is recommended to highlight the specific changes in the figures to better illustrate the differences.

**Questions:**

The results presented in this work primarily demonstrate edits on small objects or minor scene adjustments. It would be beneficial to evaluate the method’s ability to handle larger object edits within the scene, such as moving a table or a truck.

---

> ### Author Response · Authors · 2024-11-24
>
> Thanks for your informative comments. We are glad to address your concerns below. **The additional experimental results discussed below are provided at the following [anonymous link](https://imgur.com/a/lEn2N59).**
>
> **W1:it is important to acknowledge these limitations for quantitative evaluations.**
>
> We agree that quantitatively evaluating diffusion-based methods is challenging due to the potential biases, and we have included this acknowledgment in the revised limitation Sec. C.3. This is a common issue in this field, and we followed previous works (e.g., ViCA-NeRF, GaussianEditor) to conduct the user study, as well as using GPT evaluation to mitigate potential personal bias. However, such a problem cannot be fully addressed, and remains an important challenge for future research in the field
>
> In addition, to help relieve such bias in the user study, we conducted a scaled user study with **99** participants. The results, shown below (also in Figure R5), still demonstrate that we significantly outperform the baselines. We are open to adopting any further suggestions to improve the quantitative evaluations.
>
>
> |          | Instruct-NeRF2NeRF | PDS | 3DGS-Drag (Ours)|
> |------------------|:-----------------:|:-----------------:|:--------:|
> | User Preference     | 17%    | 12.7%    | **70.3%**|
>
>
>
> **W2: The qualitative figures do not clearly demonstrate the effectiveness of local editing.**
>
> Thanks for your suggestion to improve our visualization. We use a bounding box to highlight the changed region (Figure R6). Therefore, the reader can better notice which regions are changed and which are not. Such revision has also been made in Figure 5 of our revised paper. In Figure 7 of our paper, we also conducted an ablation study without using local editing, where the background was significantly changed, showing the importance of our local editing.
>
> In addition, to better illustrate the effectiveness of local editing, we calculate the similarity between the edited result’s background and the originally rendered image in the unmasked pixels. As shown in the results below, all the metrics are much better when using local editing compared to not using it. Therefore, local editing is necessary to preserve the background.
>
> |      | SSIM↑ | PSNR↑ | LPIPS↓ |
> |------------------|:-----------------:|:-----------------:|:--------:|
> | Local Editing     | **0.995**    | **43.43**    | **0.004**|
> | Non-Local Editing     | 0.90 | 24.44    | 0.158 |
>
> **Q1: the method’s ability to handle larger object edits within the scene, such as moving a table or a truck.**
>
> We are glad to provide the experiments you suggested in Figure R1. Specifically, we conduct experiments on large-scale object movements. Our tests include:
> * move the flowerpot from the center of the table to its edge,
> * move the large table,
> * move the large truck.
>
> As the results show, our method works well in both cases suggested. In addition, we can conduct long-range movements like moving the flowerpot to the side of the table. These experiments demonstrate the generalizability of our method.

---

> > ### Comment · Reviewer_Yk2w · 2024-11-26
> >
> > Most of my concerns have been addressed, and I will maintain my score.

---

> > > ### Author Response · Authors · 2024-11-26
> > > **Thanks!**
> > >
> > > Thank you for your recognition and feedback! We are glad your concerns are addressed and sincerely appreciate your constructive comments.

---

### Official Review · Reviewer_zn4P · 2024-11-02

**Soundness:** 3
**Presentation:** 3
**Contribution:** 3
**Rating:** 8
**Confidence:** 3

**Summary:**

This paper presents a method for editing gaussian-splatting 3D representation of scenes. The GS-represented 3D objects can be moved by geometrical 3D arrows designated in 3D scenes. To compensate for degradation of the rendering quality of the deformed objects, the method incorporates image correction technique based on diffusion model and  LoRA methods.

**Strengths:**

The method deals with editing of GS-represented scenes. From the demo examples, the results of the method seem reasonable.

The combination of GS-editing and diffusion-based image correction may be a practical solution.

The method seems to work better than Instruct-NeRF2NeRF.

**Weaknesses:**

Explanation of the diffusion guidance step in section 3.4 was not clear to me.  The difference from [Haque et al. 2023] is written as equation (5). Since all the information is in equation (5) and Figure 2, I'm not sure whether the proposed method is very similar to [Haque et al. 2023] or not (although there are of course differences between NeRF and GS).

**Questions:**

I would like more detailed explanation for Annealed Dataset Editing, because it should be the center of the contribution. Specifically, please explain how the method differs from Instruct-NeRF2NeRF [Haque et al. 2023].

Small comments:

Line 197: $c \in R^d$:     What is $d$?

Lines 240-241: Here, $k$ is used for two different meanings. In "top-k (k=2)"    and  as a "temporal" index in $\{p_h^k| k \n N_h^i\}$. It is misleading.

Equations (1) and (2):  Minus ("-" ) symbols are used right after sigma symbols (summation). It seems odd. I think the minus symbols normally goes right before the sigma symbols.

Figure 4: Maybe figure 4 is not referenced from the text.

---

> ### Author Response · Authors · 2024-11-24
>
> Thanks for your comments. We are happy to address your concerns below. **The additional experimental results discussed below are provided at the following [anonymous link](https://imgur.com/a/lEn2N59).**
>
> **W1: Whether the proposed method is very similar to Instruct-NeRF2NeRF or not?**
>
> We would like to clarify that our method differs significantly from Instruct-NeRF2NeRF in various aspects, beyond just the use of NeRF or 3D Gaussian Splatting. The differences are:
>
> * **Different Setting:** Instruct-NeRF2NeRF focuses on appearance and style editing, with minimal ability in geometry editing. In contrast, our method is capable of drag-based intuitive editing, which can achieve fine-grained and geometry-related editing. **As shown in the comparison in Figure 6, Instruct-NeRF2NeRF fails in these cases.**
>
> * **Different framework and techniques:** Instruct-NeRF2NeRF directly uses Instruct-Pix2Pix to update the dataset and train the NeRF progressively. In contrast, we incorporate deformation guidance and diffusion guidance, together with a multi-step drag scheduling module. All these proposed components are crucial to successfully achieving challenging intuitive drag editing in our approach.
>
> * **Difference in the diffusion process:** Instruct-NeRF2NeRF directly applies the Instruct-Pix2Pix model to obtain the edited image. However, instruct-Pix2Pix cannot consistently edit images with geometry changes, often preserving a layout similar to the original image. In contrast, our approach fine-tunes a Dreambooth model to the scene, adds noise to the deformed image, and generates the edited image from this noised version. The results show that our approach correctly maintains the geometry change from the deformed image while generating consistent content.
>
> **Q1: A more detailed explanation of Annealed Dataset Editing.**
>
> First, we would like to clarify that annealed dataset editing is one part of our contributions, not the center of the contribution. As mentioned in Lines 93-95, our primary contribution is proposing a framework for intuitive drag editing in 3D real scenes involving a deformation approach, an effective diffusion guidance technique, and a multi-step scheduling module. 3D geometry-related content editing is particularly challenging for pure diffusion-based methods like Instruct-NeRF2NeRF, whereas our method can achieve fine-grained control over these edits.
>
> Second, we would like to elaborate on the difference with Instruct-NeRF2NeRF in terms of annealed dataset editing. A qualitative ablation of using different dataset update strategies is shown in Figure R2. In Instruct-NeRF2NeRF, they edit one view each time and iteratively update the dataset. As a result, they cannot change the geometry due to inconsistent constraints from other unedited views, leading to a degenerated result in the original scene. Instead, since our method allows for consistent edits from the start, benefiting from our deformation and diffusion process, it updates all views simultaneously. To further improve the details, we perform such updates several times with the annealed strength of the diffusion model. As shown in Figure R2, the annealed dataset editing is crucial to making the edit successful.
>
>
> **Q2: Writing questions.**
>
> * Line 197 (the definition of \\(d\\)): \\(d\\) represents the dimension size for color features. Specifically, 3D Gaussian Splatting uses SH coefficients to represent color \\(c\\), enabling view-dependent effects. In practice, \\(d\\) is given by \\(d = 3 \times (\mathrm{degree}_{S} + 1)^2\\), where the SH degree \\(\mathrm{degree}_S\\) is set to 2, resulting in a dimension of 27 for \\(d\\).
> * Line 240-241 (different meaning of $k$) and the Minus symbol in Equations (1) and (2): Thanks for pointing these out, and we appreciate your corrections. We have revised them to $K$ in the current revision.
> * Figure 4 is not referenced in the text: Thanks. Figure 4 illustrates the relocated object position during the multi-step drag operation. We have revised our paper to reference it in Sec. 3.5 in the revision.

---

> > ### Comment · Reviewer_zn4P · 2024-11-26
> >
> > The feedback from the authors seems reasonable to me. I will keep the original rating because it was high from the first place.

---

> > > ### Author Response · Authors · 2024-11-26
> > > **Thanks!**
> > >
> > > Thank you for your recognition and feedback! We are glad your concerns are addressed and sincerely appreciate your constructive comments.

---

### Official Review · Reviewer_zAUe · 2024-11-03

**Soundness:** 3
**Presentation:** 3
**Contribution:** 3
**Rating:** 6
**Confidence:** 3

**Summary:**

This paper introduces a drag-based 3D editing framework for 3D Gaussian representations. The approach employs deformation guidance to deform 3D Gaussians (3DGS) from a specified handle point to a target point, followed by diffusion guidance to enhance visual quality. Experimental results demonstrate that this method outperforms existing baselines.

**Strengths:**

1. The paper presents an innovative drag-based approach for editing 3DGS.

2. The experiments are thorough and provide convincing evidence of the method’s effectiveness.

3. The paper is well-structured.

**Weaknesses:**

1. In the case where the wall is widened (bottom right in Fig. 1), there is a noticeable color discrepancy between the original and the widened sections. Additionally, leaves near the edited boundary on the wall appear blurry.

2. In the baseline comparison, authors claim that "there’s no exact previous work on intuitive 3D drag operation". However, this is inaccurate.  ARSP [1] implements drag-based 3D editing using mesh-based deformation techniques. Interactive3D [2] offers a set of deformable 3D point operations on 3DGS and utilizes SDS to optimize the deformed 3DGS.

[1] Yoo, Seungwoo, et al. "As-Plausible-As-Possible: Plausibility-Aware Mesh Deformation Using 2D Diffusion Priors." Proceedings of the IEEE/CVF Conference on Computer Vision and Pattern Recognition. 2024.

[2] Dong, Shaocong, et al. "Interactive3D: Create What You Want by Interactive 3D Generation." Proceedings of the IEEE/CVF Conference on Computer Vision and Pattern Recognition. 2024.

**Questions:**

1. Can this method reposition the entire flowerpot (as in Fig. 2) rather than only elongating it?

**Details Of Ethics Concerns:**

No ethical concerns have been identified.

---

> ### Author Response · Authors · 2024-11-24
>
> Thanks for your encouraging comments, and we are happy to address your concerns below. **The additional experimental results discussed below are provided at the following [anonymous link](https://imgur.com/a/lEn2N59).**
>
> **W1.1: There is a noticeable color discrepancy between the original and the widened sections.**
>
> We conduct a study on this in Figure R4. Initially, we use a small radius for handle points, which leads to noticeable gray artifacts on the boundary of the wall. The color discrepancy happens because that gray part is recognized as a potential color shift from the diffusion model (Figure R4 (a)). Thus, although the edited result can correctly model the wall, the color is shifted. We also find that **such a problem can be simply addressed by changing to a larger radius to avoid artifacts on the boundary** (Figure R4 (b)). Since the wall can be correctly modeled in both cases, we believe that our approach remains robust and effective without introducing noticeable artifacts. The detailed texture is generated through diffusion guidance, while our intuitive dragging operation provides the flexibility to refine the results at a fine-grained level.
>
> **W1.2: The leaves near the edited boundary on the wall appear blurry.**
>
> The blurry appearance of leaves near edited boundaries stems from their high-frequency nature, which presents a significant challenge for existing 3D editing methods. As shown in Figure 6 of the main paper, our approach achieves notably better sharpness and clarity in these areas than baseline methods.
>
> **W2: Inaccurate claim when comparing baselines.**
> Thanks for pointing this out, and we apologize for this inaccurate and confusing claim. We have modified our claim to “there is no directly comparable work on intuitive 3D drag operation in real scenes” in the revised paper (Line 467). We also revised our paper to reference these two related works in Lines 146-147.
>
> Compared with the mentioned papers, we have the following important differences:
>
> **Different settings**: Both Interactive3D and APAP focus on single object editing without background, but our work focuses on large real scenes, where we need to deal with challenges like photorealistic objects, complex backgrounds, lightning, and inaccurate geometric modeling. Thus, these methods cannot be applied in our setting, since they are designed for single objects with well-captured geometry.
>
> **Technical difference**: Both Interactive3D and APAP follow the paradigm of using the Score Distillation Sampling (SDS) loss for editing. Such a loss is initially proposed for single object generation. In contrast, our approach introduces a deformation-and-correction paradigm to address the challenges of photorealistic editing in real scenes.
> In summary, these object-centric drag editing methods are not applicable for comparison in our setting.
>
> **Q1: Can this method reposition the entire flowerpot (as in Fig. 2) rather than only elongating it?**
>
> Thanks for your constructive question. Yes, our method can successfully reposition the entire flowerpot. As shown in Figure R1, we can reposition the flowerpot from the center of the table to its edge.

---

> > ### Comment · Reviewer_zAUe · 2024-11-26
> >
> > Most of my concerns are addressed. I will keep my score.

---

> > > ### Author Response · Authors · 2024-11-26
> > > **Thanks!**
> > >
> > > Thank you for your recognition and feedback! We are glad your concerns are addressed and sincerely appreciate your constructive comments.

---

### Official Review · Reviewer_SwqC · 2024-11-04

**Soundness:** 3
**Presentation:** 3
**Contribution:** 3
**Rating:** 6
**Confidence:** 2

**Summary:**

This paper proposes a drag-based 3D Gaussian editing method. The pipeline is divided into two steps. It first binds 3D Gaussians to nearby handle points, and copy-and-paste the handle points to the target position. Secondly, it leverage a diffusion model to correct artifacts caused by Gaussian movements. Specifically, it converts the rendered image to sketch level to help the diffusion model understand the complete the deformed part. It also use Dreambooth and iterative dataset updating to better ensure 3D consistency. The paper also extends from one-step editing to multi-step editing, providing a stable and reliable 3D Gaussian editing method.

**Strengths:**

This paper proposes a drag-based 3D Gaussian editing method, while previous methods focus on text-guided editing. It uses a fine-tuned diffusion model to provide view-consistent correction to the edited scene and unsure rendering quality through iterative dataset updates. It also introduces multi-step drag editing to allow long-distance editing operations.

**Weaknesses:**

The examples shown in the paper have small movements. Will the method fail if we move a large object to a large range of movement? I suggest the author to provide some failure cases, which will better illustrate the upper bound of the method and make the work more solid.

**Questions:**

No.

---

> ### Author Response · Authors · 2024-11-24
>
> Thanks for reviewing our paper and the constructive comments. We improve our paper based on your suggestions and address your concerns below. **The additional experimental results discussed below are provided at the following [anonymous link](https://imgur.com/a/lEn2N59).**
>
> **W1.1: Will the method fail if we move a large object to a large range?**
>
> In most cases, our method will succeed. We conduct experiments on large-scale object movements in Figure R1. Our tests include:
>
> * move the flowerpot from the center of the table to its edge,
>
> * move the large table,
>
> * move the large truck.
>
> As the results show, our method works well in each case.
>
> Meanwhile, we have identified limitations when moving objects from the background to the foreground (Figure R3 (a)). This occurs primarily because the background objects are initially partially observed and not sufficiently modeled in 3D. Thus, they do not have correct rendering results for other views when dragged to the center. We have also revised the paper to include this in our limitation section (Sec. C.1).
>
> **W1.2: Include more failure cases.**
>
> In addition to the background modeling, our discussion of failure cases also includes the boundary region issue in Figure R3 (b). When drag operations partially extend beyond most cameras' visible area, optimization becomes challenging due to limited edited image availability. In our original submission, the limitation section also discussed such limitations (now it is Appendix Sec. C.2 in the revised paper). We are happy to discuss this if you have additional comments.

---

### Author Response · Authors · 2024-11-24
**General Response to All**

We are thankful for the feedback and suggestions from all the reviewers. We are glad that the reviewers recognize our novel drag-based approach for editing 3D Gaussian representations (SwqC, zAUe, Yk2w), the effective combination of deformation guidance and diffusion-based correction that ensures view-consistent results (SwqC, zn4P, Yk2w). The reviewers particularly appreciate our thorough experimental validation (zAUe, Yk2w) and the practical solution our method provides compared to existing baselines (zAUe, zn4P, Yk2w). It is our pleasure to see that our paper is well-structured and easy to follow (zAUe, Yk2w).

We address each of the reviewers' concerns in the individual responses below. We have also revised our paper based on their comments, and the updated parts are highlighted in $\text{\textit{\color{orange}{orange}}}$. For the convenience of checking, here we present the additional experimental results and illustrations in an [anonymous link](https://imgur.com/a/lEn2N59), which include:
* Additional results on large movements and large objects.
* Ablation on our dataset editing strategy.
* Additional study on failure cases of our method.
* Analysis of the “extending the wall” edit.
* A larger-scale user study with 99 participants.
* Quantitative ablation on the effectiveness of local editing.
* Highlighted qualitative results with bounding boxes.

We look forward to your further comments and suggestions.

---

### Meta-Review · Area_Chair_iWq1 · 2024-12-20

**Metareview:**

The paper presents an intuitive and effective drag-based method for manipulating scenes in 3D Gaussian Splatting (3DGS). The proposed approach is novel, utilizing deformation guidance to deform 3DGS and diffusion guidance to correct artifacts caused by Gaussian movement. The experiments are comprehensive and provide convincing results. The primary strength of the paper lies in its novelty in the deformation-and-correction strategy. The reviews mentioned some previous work on intuitive 3D drag operations. While the proposed method has differences, it would be better to add the references and discussion in the revised paper. Nevertheless, the paper makes a valuable contribution by proposing a novel method for intuitive drag-based 3DGS manipulation.

**Additional Comments On Reviewer Discussion:**

The paper mainly received positive feedback in the initial reviews, with some issues addressed in the rebuttal.

One reviewer noted that some prior works also facilitate intuitive 3D drag operations. The rebuttal clarified that these approaches differ from the proposed method in terms of settings (objects vs. scenes) and techniques (Score Distillation Sampling vs. deformation-and-correction). Additionally, the rebuttal explained how the proposed method differs from Instruct-NeRF2NeRF.

Reviewers questioned how well the method performs for larger motions. The rebuttal provided additional results demonstrating the method's effectiveness on large movements and objects. It also discussed failure cases, offering insights into the approach's limitations.

Some reviewers observed artifacts in specific examples. The rebuttal addressed this by explaining the causes of these artifacts and suggesting potential solutions to mitigate them.

A reviewer expressed concerns about the number of participants in the user study. The rebuttal addressed this by conducting a larger-scale user study, strengthening the validity of the results.

The rebuttal effectively addressed most of the concerns raised during the review process. All reviewers were optimistic about the paper by the end of the discussion stage.

---

### Decision · Program_Chairs · 2025-01-22

Accept (Poster)